# Basin-scale Evaluation of the Noah-MP Land Surface Model for Runoff and Snow Generation in the Missouri River Basin: Insights and Recommendations for Parameterization Scheme Selection

5 Eunsaem Cho<sup>1,2</sup>, Eunsang Cho<sup>3</sup>, Carrie Vuyovich<sup>1</sup>, Bailing Li<sup>1,2</sup>, Jennifer M. Jacobs<sup>4,5</sup>

<sup>1</sup>Hydrological Sciences Laboratory, NASA Goddard Space Flight Center, Greenbelt, MD, USA <sup>2</sup>Earth System Science Interdisciplinary Center, University of Maryland, College Park, MD, USA

<sup>3</sup>Ingram School of Engineering, Texas State University, San Marcos, TX, USA

<sup>4</sup>Department of Civil and Environmental Engineering, University of New Hampshire, Durham, NH, USA

5 Earth Systems Research Center, Institute for the Study of Earth, Oceans, and Space, University of New Hampshire, Durham, NH, USA

Correspondence to: Eunsang Cho (eunsang.cho@txstate.edu)

**Abstract.** Process-based land surface models, such as the Noah-Multiparameterization (Noah-MP) model, are widely used for large-scale hydrologic simulations because of their flexibility in selecting multiple parameterization schemes. However, limited guidance on choosing appropriate configurations constrains their reliability in representing runoff

and snowmelt dynamics across diverse land-cover and snow conditions. This study evaluates the default parameterization scheme and four alternative parameterization schemes in the Noah-MP land surface model, including Runoff and Groundwater (RUN), Surface Exchange Coefficient for Heat (SFC), Frozen Soil Permeability (INF), and Snow/Soil Temperature Time Scheme (STC), across 50 Hydro-Climate Data Networks (HCDNs) in the Missouri

River Basin. Model performance was evaluated using USGS streamflow observations and snow water equivalent (SWE) estimates from the University of Arizona dataset for 2014 to 2023. Results showed that the alternative schemes generally improved runoff simulation compared to the default scheme through better representing key hydrological and thermodynamic processes. Specifically, the RUN, SFC, INF, and STC experiments improved the Kling–Gupta

Efficiency (KGE) by 0.19, 0.37, 0.48, and 0.14, respectively in representative subbasins, through enhanced groundwater dynamics, reduced evapotranspiration bias, improved rapid runoff response, and more accurate SWE evolution. SWE evaluation further indicates that the STC experiment reduced the mean bias of the April–July runoff-

to-maximum SWE ratio by 12–32% in high-elevation subbasins, reflecting improved representation of snowmelt-driven runoff. These results highlight the importance of basin-specific parameterization schemes within Noah-MP to improve hydrological prediction and water management across diverse hydroclimatic regions. The findings further

indicate optimal parameterization schemes for different climates, land cover, and snow regimes.

# 1 Introduction

Snowmelt is one of the most important hydrological processes in snow-dominated regions, as it governs the timing and magnitude of water availability for ecosystems, agriculture, and human consumption (Barnett et al., 2005; Qin et al., 2020; Siirila-Woodburn et al., 2021; Cho et al., 2023). The gradual release of snowmelt sustains streamflow during

dry seasons, replenishes groundwater aquifers, and supports downstream water supplies (Huntington and Niswonger, 2012; Carroll et al., 2019). Conversely, rapid or excessive snowmelt can trigger flooding, infrastructure damage, and cascading hazards (Vormoor et al., 2016; Barnhart et al., 2020). Capturing these dynamics requires a robust understanding of hydrological processes—including snow accumulation, energy balance, infiltration, and runoff generation—because they determine how snowmelt is translated into streamflow.

Over the past several decades, snowmelt-related flooding has repeatedly affected communities across the United States and Canada, further demonstrating the widespread risks associated with changes in snowpack and runoff regimes (Graybeal et al., 2006; Stadnyk et al., 2016; Cho et al., 2021). The devastating 2019 Midwest Spring Flood exemplified snowmelt's societal impact, as rain-on-snow conditions breached more than 50 levees, unleashed unprecedented 45 Missouri River Basin damages, and amassed nearly \$12 billion in losses (NCEI, 2020; Francis & Melliger, 2021). Such events underscore the vulnerability of agricultural, urban, and ecological systems to changes in snowpack dynamics and runoff generation, highlighting the need for robust modeling frameworks capable of capturing both seasonal snowpack accumulation and the rapid hydrological transitions associated with snowmelt and rain-on-snow events (Barnett et al., 2005; Musselman et al., 2018).

Process-based land surface models (LSMs) have been widely employed for operational flood forecasting and water supply predictions (Alfieri et al., 2013; Wood & Lettenmaier, 2006; Cosgrove et al., 2024). LSMs simulate coupled exchanges of energy, water, and carbon between the land and atmosphere, enabling both climate diagnostics and hydrological forecasting (Overgaard et al., 2006; Abramowitz et al., 2008; Fisher and Koven, 2020). The Noah with multi-parameterization options (Noah-MP) model is one of the most widely adopted frameworks due to its role as the hydrologic core of the U.S. National Water Model and its unique design that allows multiple parameterization options for key processes including snow physics, runoff generation, frozen soil dynamics, and canopy exchange (Niu et al., 2011; Cuntz et al., 2016; He et al., 2023; Cosgrove et al., 2024). Theoretically, this flexibility enables Noah-MP users to select schemes that best represent local environmental conditions.

In practice, however, most applications of Noah-MP rely on its default physics or employ a single parameterization scheme across large domains (Ma et al., 2017; Li et al., 2022; Yin et al., 2023). Sensitivity studies have shown that snowpack accumulation, soil freeze-thaw dynamics, and runoff generation are highly dependent on parameterization choices (You et al., 2020; Cao et al., 2024), yet guidance for selecting among schemes remains limited. Prior evaluations have tended to focus on global- or continental- scale domains, emphasizing broad performance rather than basin-specific applicability (Gan et al., 2019; You et al., 2020). Moreover, while some studies have assessed SWE simulation (You et al., 2020; Yang et al., 2023) or runoff calibration (Su et al., 2024) independently, comprehensive analyses linking snow dynamics and runoff processes across diverse basin settings remain scarce. This knowledge gap constrains the ability of modelers to optimize Noah-MP for snow-dominated basins where both water supply forecasting and flood risk reduction depend on accurate snowmelt representation.

Building on this context, this study focuses on systematically evaluating four alternative Noah-MP parameterization schemes governing runoff and groundwater, surface exchange coefficients, frozen soil permeability, and snow/soil temperature dynamics across 50 minimally regulated subbasins across the Missouri River Basin. The objectives of this study are threefold: (1) to quantify how Noah-MP parameterization schemes influence runoff simulation in the Missouri River Basin; (2) to evaluate how basin characteristics—including climate class, land cover, and snow regime—control parameterization effectiveness; and (3) to assess how alternative schemes improve SWE and snowmelt-driven runoff representation relative to default model settings. To achieve these objectives, we introduce the Missouri River Basin as the target study area and describe our model design and experimental setup in Section 2.

Section 3 outlines the datasets and evaluation methods used for runoff and SWE comparisons. The results of parameterization experiments are presented in Section 4, followed by a discussion of region-specific sensitivities, implications for scheme selection, and broader modeling challenges in Section 5. Section 6 provides concluding remarks, highlighting recommendations for Noah-MP parameterization schemes and future directions for improving snow and runoff simulation in land surface models.

### 85 2 Study Area

This study focuses on the Missouri River Basin (USGS hydrologic unit code HUC-2 10) and Hydro-Climate Data Network (HCDN) located in Missouri River Basin (Fig. 1a). This basin was selected because it is highly vulnerable to spring snowmelt floods, as demonstrated by the devastating 2019 Midwest Spring Flood (Francis and Melliger, 2021; Shirzaei et al., 2021; Velasquez et al., 2023). In addition, the Missouri River Basin climate ranges from alpine tundra in the Rocky Mountains, through semi-arid steppe across the Great Plains, to humid subtropical conditions in the eastern lowlands (Qiao et al., 2014; Wise et al., 2018). In Fig. 1b, the digital elevation model (DEM) highlights the Rocky Mountain region in western part of the basin, which plays a dominant role in snow accumulation and snowmelt-driven runoff in the Missouri River Basin. Land cover type also varies, transitioning from high-elevation evergreen forests to wide grasslands and croplands (Jordan et al., 2012; Ahiablame et al., 2017). Snow regimes are diverse, with tundra, boreal, and montane forest snow in the Rockies, prairie snow in the Great Plains, and ephemeral snow in the southern lowlands (Sturm and Liston, 2021). The combination of varied climate and land cover and substantial snow make the Missouri River Basin ideal for evaluating Noah-MP parameterization schemes relevant to snowmelt-driven runoff.

Figure 1 Overview of the Missouri River Basin showing (a) the 50 selected USGS Hydro-Climatic Data Networks (HCDNs) with at least 25 years of record and minimal dam influence, where numbers indicate site identifiers. Maps illustrating (b) the DEM, (c) climate classification, (d) land cover, and (e) snow classification. Location of selected subbasins for (f) runoff and (g) SWE comparisons. Grey shading represents basin boundaries.

For runoff and SWE evaluation, we initially identified 75 subbasins with the USGS streamflow gauges in HCDN, which are characterized by minimal anthropogenic influence (USGS, 2012). Among these, 64 subbasins had at least 25 years of continuous observation, ensuring gauge reliability and operational consistency. Additionally, we excluded

14 subbasins containing dams with controlled outflows, as identified in the National Inventory of Dams (USACE, 2021), to avoid potential impacts of dam operations on natural streamflow patterns. In total, 50 subbasins were selected for analysis, and the upstream drainage area for each gauge was delineated using DEM of the Missouri River Basin (Fig. 1).

#### 3 Data

In this study, land cover, climate and snow classification were used to characterize the basins. The land cover type for each subbasin was determined using the University of Maryland Global Land Cover product (Hansen et al., 2000; https://ldas.gsfc.nasa.gov/nldas/vegetation-class), which serves as the vegetation classification dataset for the North American Land Data Assimilation System (NLDAS) at 0.125-degree resolution. Climate classification was assigned using the Köppen-Geiger climate classification map (Beck et al., 2018; https://www.gloh2o.org/koppen/) at a 1-km spatial resolution. The predominant snow class within each subbasin was identified using the Sturm snow classification map (Sturm and Liston, 2021; https://nsidc.org/data/nsidc-0768/versions/1), also available at 1-km resolution, which characterizes snow accumulation and ablation patterns based on long-term records of air temperature, precipitation, wind speed, and land-cover.

For model evaluation, we used streamflow observations from USGS HCDN gauges to evaluate simulated runoff and the University of Arizona (UA) gridded SWE dataset (Broxton et al., 2019) to evaluate SWE simulation. The observation-based UA SWE combines station-based SWE and snow depth observations from the Snowpack Telemetry (SNOTEL) network and National Weather Service Cooperative Observer Program (COOP) network with modeled SWE based on an empirical temperature index snow model. Daily gridded UA SWE values at 1 km from January 2014 to September 2023 were utilized in this study.

### 4 Model and Methods

### 130 4.1 Noah-MP

Noah-MP is a widely used, community-driven model designed to simulate the coupled energy, water, and carbon cycles at the land surface (Niu et al., 2011; Yang et al., 2011). Developed as an enhancement to the original Noah LSM, Noah-MP incorporates multiple parameterization options for vegetation processes, canopy radiative transfer, multi-layer snow physics, soil thermal and moisture dynamics, and groundwater interactions (Niu et al., 2011; Yang et al., 2011; He et al., 2023). The availability of multiple parameterization options within Noah-MP is particularly valuable for regional applications, because it allows researchers and practitioners to optimize model performance by selecting the most appropriate physical representations for specific geographical and climatological conditions (Hong et al., 2014; Ma et al., 2017; He et al., 2023).

In this study, Noah-MP 4.0.1 was forced with meteorological data from NLDAS-2. NLDAS-2 provides high-quality, observation-based atmospheric forcing at hourly temporal resolution and 0.125° (~12 km) spatial resolution, spanning

from 1979 to the present (Xia et al., 2012). The forcing dataset includes hourly integrated precipitation, 2-meter air temperature and specific humidity, incoming shortwave and longwave radiation, 10-meter zonal and meridional wind speed, and surface pressure, making it well suited for driving land surface and hydrological models (Xia et al., 2012; Xia et al., 2014; Nearing et al., 2016). NLDAS-2 has been extensively validated and is widely used in hydrological and climate studies across North America, supporting applications such as streamflow simulation, drought monitoring, and water resource management (Xia et al., 2012; Hao et al., 2016; Xia et al., 2016; Pascolini-Campbell and Reager, 2024).

Initial conditions for all experiments were established through a spin-up procedure. The spin-up simulation was run from January 1, 1980 to December 31, 1999, using NLDAS-2 forcing data with all default parameterization scheme settings. Following the spin-up, the model states at the end of 1999 were used as initial conditions for all subsequent experimental runs, providing a consistent baseline for comparative analysis across different parameterization configurations and sensitivity experiments.

### 55 4.2 Parameterization Schemes for Snow-dominated Basin

To assess Noah-MP's sensitivity to parameterization schemes, this study evaluates four alternative schemes governing runoff and groundwater generation, surface energy exchange, frozen soil permeability, and subsurface thermal dynamics (Table 1). These schemes were selected based on their perceived influence on snowmelt partitioning, groundwater interactions, and flood generation in prior sensitivity studies (Niu et al., 2011; You et al., 2020; Li et al., 2022; Yang et al., 2023). In the following, we briefly discuss these four schemes and their impact on snow and runoff simulations.

Table 1 Default and alternative parameterization schemes for the four Noah-MP experiments.

| Experiment | Physical process             | Default scheme       | Alternative scheme      |  |  |
|------------|------------------------------|----------------------|-------------------------|--|--|
|            |                              | TOPMODEL with        | TOPMODEL with an        |  |  |
| RUN        | Runoff and Groundwater       | groundwater storage  | equilibrium water table |  |  |
|            |                              | (Niu et al., 2007)   | (Niu et al., 2005)      |  |  |
| SFC        | Surface Exchange Coefficient | Monin-Obukhov        | Chen97                  |  |  |
|            | for Heat                     | (Brutsaert, 1982)    | (Chen et al., 1997)     |  |  |
| D.T.       | E 0.1D 1.11                  | NY06                 | Koren99                 |  |  |
| INF        | Frozen Soil Permeability     | (Niu and Yang, 2006) | (Koren et al. 1999)     |  |  |
| STC        | Snow/Soil Temperature Time   | Semi-implicit        | Full implicit           |  |  |
|            | Scheme                       | (Niu et al., 2011)   | (Ek et al., 2003)       |  |  |

### 65 4.2.1 Runoff and Groundwater Process

For the runoff and groundwater simulations, we evaluated two options (1) TOPMODEL with groundwater storage and (2) TOPMODEL with the equilibrium water table. Both schemes compute surface and subsurface runoff as functions of the water table depth but they differ in how the water table depth is determined.

In TOPMODEL with groundwater storage scheme (Niu et al., 2007), the water table depth is estimated using a linear reservoir located beneath the unsaturated soils. Groundwater storage in this unconfined aquifer is dynamically updated at each time step, reflecting the net gain or loss between groundwater recharge and discharge:

$$\frac{\partial d_{wt}GWS}{\partial t} = Q_{recharge} - Q_{discharge} \tag{1}$$

where GWS represents groundwater storage changes;  $Q_{recharge}$  is groundwater recharge, calculated using Darcy's law (Darcy, 1856);  $Q_{discharge}$ , groundwater discharge, represents subsurface runoff estimated based on the soil hydraulic characteristics and topographic properties. Changes in the groundwater table depth are then calculated by dividing groundwater storage changes with the specific yield, which is set at 0.2 globally in Noah-MP (Niu et al., 2007).

TOPMODEL with the equilibrium water table scheme estimates water table depths using a conceptual soil column, following the approach of Niu et al. (2005). The initial water table depth is set to be three times the default Noah-MP soil column depth (the original coarse soil column). The model also assumes a hypothetical, finer, and deeper soil column in addition to the original one. Total water deficit is calculated for the original soil column using maximum soil moisture and liquid water content, and for the hypothetical deeper soil column using the Clapp-Hornberger relation (Clapp and Hornberger, 1978), which incorporates saturated soil matrix potential and a soil-type exponent. The equilibrium water table depth is determined by iteratively adjusting the water table depth until the difference in total water deficit between the two approaches is less than 0.01 m.

# 190 4.2.2 Surface Exchange Coefficient for Heat

For the process of the surface exchange coefficient for heat, we considered (1) the Monin-Obukhov (M-O) scheme (Brutsaert, 1982) and (2) the Chen97 scheme (Chen et al., 1997). The M-O scheme is based on Monin-Obukhov similarity theory, while Chen97 scheme (used in Noah version 3.0) differs in how surface roughness and displacement height are treated.

The M-O scheme uses the following equation for the surface exchange coefficient for heat  $(C_H)$ :

$$C_{H} = \frac{\kappa^{2}}{\left[\ln\left(\frac{z-d_{0}}{z_{0m}}\right) - \psi_{m}\left(\frac{z-d_{0}}{z_{0}}\right)\right] \left[\ln\left(\frac{z-d_{0}}{z_{0h}}\right) - \psi_{h}\left(\frac{z-d_{0}}{L}\right)\right]}$$
(2)

where  $\kappa$  is the von Kármán constant; z is the reference height;  $d_0$  is the zero-displacement height;  $z_{0m}$  and  $z_{0h}$  are the roughness lengths for momentum and heat (assumed to be the same in M-O scheme); L is the Monin-Obukhov length, and  $\psi_m$  and  $\psi_h$  are stability correction functions for momentum and heat, respectively.

The Chen 97 scheme uses a slightly different approach to compute  $C_H$ :

$$C_{H} = \frac{\kappa^{2}}{\left[\ln\left(\frac{z}{z_{0m}}\right) - \psi_{m}\left(\frac{z}{L}\right) + \psi_{m}\left(\frac{z_{0m}}{L}\right)\right] \left[\ln\left(\frac{z}{z_{0h}}\right) - \psi_{h}\left(\frac{z}{L}\right) + \psi_{h}\left(\frac{z_{0h}}{L}\right)\right]}$$
(3)

This scheme explicitly distinguishes between the roughness lengths for momentum  $(z_{0m})$  and heat  $(z_{0h})$  where the latter is determined based on the Reynolds number using an empirical equation.

The key difference is that the M-O scheme can drive  $C_H$  toward near-zero under very stable (e.g., nighttime, weak-wind) conditions, whereas the Chen97 scheme relaxes this suppression and retains a small but finite exchange. As a result, Chen97 scheme often yields a higher  $C_H$  than the M-O scheme (Niu et al., 2011). A higher  $C_H$  enhances turbulent ventilation and sensible heat exchange, which cools the land surface and reduces the vapor pressure gradient, thereby suppressing ET, particularly under dry or energy-limited conditions.

# 215 4.2.3 Frozen Soil Permeability

For simulating soil water diffusivity under frozen conditions, we examined Niu and Yang (2006, NY06) scheme and Koren et al (1999, Koren99) scheme. The approaches differ in their physical assumptions and mathematical equation for soil water diffusivity ( $\lambda$ ).

In the NY06 scheme, soil water diffusivity,  $\lambda_{NY}$ , is calculated by

$$\lambda_{NY} = \left(1 - f_{imp,soil}\right) \times \lambda_{sat} \times \left[\max\left(0.01, \frac{\theta_{soil}}{\theta_{soil,max}}\right)\right]^{B_{exp}+2} \tag{4}$$

where  $\lambda_{sat}$  is the saturated soil water diffusivity,  $f_{imp,soil}$  is the impermeable fraction of the soil layer,  $B_{exp}$  is a soil texture-dependent exponent,  $\theta_{soil}$  is the actual soil moisture, and  $\theta_{soil,max}$  is the maximum soil moisture. This equation produces a smooth reduction in diffusivity as soil freezing progresses, allowing water movement to remain possible even in partially frozen soils.

In the Koren99 scheme, soil water diffusivity,  $\lambda_{Kor}$ , is calculated as

$$\lambda_{Kor} = \lambda_{sat} \times \left[ \max \left( 0.01, \frac{\theta_{soil}}{\theta_{soil,max}} \right) \right]^{B_{exp}+2}$$
 (5)

for unfrozen soils and for frozen soils, it is computed as

$$\lambda_{Kor} = \left[\frac{1}{1 + (500 \times W_{ice,soil})^3}\right] \times \lambda_{sat} \left[\max\left(0.01, \frac{\theta_{soil}}{\theta_{soil,max}}\right)\right]^{B_{exp}+2} + \left[1 - \frac{1}{1 + (500 \times W_{ice,soil})^3}\right] \times 235 \quad \lambda_{sat} \left[\min\left(\frac{0.05}{\theta_{soil,max}}, \max\left(0.01, \frac{\theta_{soil}}{\theta_{soil,max}}\right)\right)\right]^{B_{exp}+2}$$
(6)

where  $W_{ice,soil}$  is the volumetric ice content. This nonlinear formulation causes a much sharper reduction in diffusivity as ice content increases, reflecting reduced water movement in frozen soils (Niu et al., 2011).

### 4.2.4 Snow/Soil Temperature Time Scheme

The semi-implicit (Niu et al., 2011) and full implicit (Ek et al., 2003) schemes are two approaches for simulating the evolution of snow and soil temperature in Noah-MP. Each scheme enforces that ground temperature beneath snow cannot exceed the freezing point, but they differ in how this constraint and dynamic processes are handled. In the semi-implicit scheme, the model checks during each timestep whether the snow depth exceeds 0.05 meters and if the ground temperature ( $T_g$ ) is above the freezing point ( $T_{frz}$ ). If both conditions are met, the ground temperature is immediately reset to the freezing point.

This adjustment is implicit because it enforces the phase change constraint within the timestep, ensuring that the ground cannot physically be warmer than the freezing point while snow is present. The process is dynamic because it is performed at every timestep, allowing the scheme to respond to evolving snow and temperature conditions as the simulation progresses. After the temperature is reset, net longwave radiation, sensible heat, latent and ground heat fluxes are recalculated using the updated ground temperature, ensuring that the energy balance reflects new physical states. The semi-implicit method thus maintains the phase change constraint and energy conservation dynamically, without requiring fully coupling all energy and mass fluxes at once.

The full implicit scheme does not recalculate fluxes but applies the freezing constraint separately to vegetated and bare soil fractions. If the snow depth is higher than 0.05 m and the ground temperature is above freezing point, both the vegetated ground temperature ( $T_{g,veg}$ ) and bare ground temperature ( $T_{g,bare}$ ) are set to the freezing point. Here, bare ground temperature is used to calculate the surface temperature, which represents the skin temperature of the land surface and influences snowmelt and SWE accumulation through surface energy balance processes.

The semi-implicit and full implicit schemes both enforce the constraint that ground temperature beneath snow cannot exceed the freezing point, but they differ in how surface temperature is determined. In the semi-implicit scheme, surface temperature is consistently derived from flux—temperature relationships. In contrast, the full implicit scheme computes the grid-mean surface temperature as a weighted average of bare ground temperature and canopy temperature for the vegetated fraction. This structural difference may cause the semi-implicit scheme to produce more abrupt SWE changes due to the discrete flux recalculation, whereas the full implicit scheme can capture more spatially variable SWE evolution by explicitly representing heterogeneous surface fractions.

#### 4.3 Performance Evaluation with Statistical Metrics

In this study, Kling-Gupta Efficiency (KGE) is employed as the primary performance metric to evaluate Noah-MP hydrological simulation. The KGE metric was developed to address limitations of traditional performance measures such as Nash-Sutcliffe Efficiency, and has been widely adopted for calibration and evaluation of hydrological models (Gupta et al., 2009; Liu et al., 2022; Cinkus et al., 2023). KGE is calculated as:

$$KGE = 1 - \sqrt{(r-1)^2 + (\beta - 1)^2 + (\gamma - 1)^2}$$
(7)

where r represents the linear correlation coefficient between observed and simulated values,  $\beta$  is the ratio of simulated to observed means, and  $\gamma$  is the ratio of simulated to observed standard deviations. KGE ranges from negative infinity to 1, with values close to 1 indicating better model performance. Knoben et al. (2019) showed that KGE = -0.414 represents the performance of prediction using the observation mean.

### 280 5 Results

### 5.1 Effects of Parameterization Schemes on Water Balance components in the Missouri Basin

To evaluate the influence of parameterization schemes, a water balance analysis was conducted over the entire Missouri River Basin domain. All  $0.125^{\circ}$  resolution grid cells within the basin were considered. The analysis was performed using daily values from 2014 to 2023, with water years defined as October 1 through September 30 of the following year. For each water year, the following components were computed: total precipitation (P), total evapotranspiration (ET), change in groundwater storage  $(\Delta SM)$ ; groundwater storage at the final day minus the first day of the year), change in soil moisture storage  $(\Delta SM)$ ; soil moisture at final day minus initial day), surface runoff  $(Q_{s})$ , subsurface runoff  $(Q_{sb})$ , and total runoff  $(Q_{total})$ ; surface runoff + subsurface runoff). These variables were aggregated spatially across the basin and temporally averaged (sum for fluxes) over a 10-year period to represent long-term hydrologic behavior under each parameterization scheme.

Table 2 summarizes the 10-year mean values of each water balance component across the five parameterization schemes evaluated. Mean annual precipitation for the five experiments was 535.4 mm, of which 464–480 mm (87–90%) was lost through evapotranspiration and 65–81 mm (12–15%) was routed as total runoff. Variability among experiments was mainly expressed in the ET and runoff, with ET differences of up to 16 mm balanced by corresponding changes in runoff. For all experiments, the water-budget residual—defined as Residual =  $P - ET - \Delta GW - \Delta SM - Q_{total}$ —was below 1%, indicating tight water-balance closure in the simulations.

**Table 2** Basin-averaged, 10-year mean water balance components across five parameterization schemes for the entire Missouri 300 River Basin. Components include precipitation (P), evapotranspiration (ET), change in groundwater storage ( $\Delta GW$ ), change in soil moisture ( $\Delta SM$ ), surface runoff ( $Q_s$ ), subsurface runoff ( $Q_{sb}$ ), and total runoff ( $Q_{total}$ ).

| Experiment | P (mm)         | ET (mm)        | △GW (mm)     | △SM (mm)     | $Q_s$ (mm)   | $Q_{sb}$ (mm) | $Q_{total}$ (mm) |
|------------|----------------|----------------|--------------|--------------|--------------|---------------|------------------|
| Default    | 535.4          | 479.7          | -0.3         | -6.1         | 20.9         | 44.0          | 64.8             |
| RUN        | 535.4          | 473.7          | 2.8          | -5.6         | 27.2         | 41.1          | 68.3             |
| SFC        | 535.4          | 463.6          | 0.1          | -6.3         | 25.2         | 56.0          | 81.2             |
| INF<br>STC | 535.4<br>535.4 | 476.7<br>474.1 | -0.1<br>-0.3 | -6.0<br>-5.8 | 28.2<br>22.0 | 39.4<br>48.3  | 67.6<br>70.3     |

Table 2 demonstrates how key hydrological components differed by Noah-MP model parameterization scheme. The RUN experiment, which assumes an equilibrium water table, produced the largest increase in groundwater storage  $(\Delta GW = +2.8 \text{ mm})$ . The SFC experiment resulted in the lowest evapotranspiration (ET = 463.6 mm), because its alternative scheme limits latent heat flux. This reduction in ET contributed to the highest total runoff ( $Q_{total} = 81.2 \text{ mm}$ ), largely through increased subsurface runoff. In contrast, the INF experiment yielded the highest surface runoff ( $Q_s = 28.2 \text{ mm}$ ), likely reflecting reduced infiltration capacity under frozen soil conditions. The increased surface runoff was partially offset by lower subsurface flow, maintaining a total runoff value comparable to other schemes. The STC experiment produced higher surface, subsurface and total runoff than the default configuration, reflecting altered snow dynamics under its alternative snow/soil temperature time scheme.

# 5.2 Evaluation of Runoff Simulation across Parameterization Schemes

Runoff simulation performance was evaluated by comparing monthly Noah-MP total runoff with observed streamflow at the 50 USGS HCDNs (Fig. 2). Overall, runoff simulation performed well in high-elevation area in west of the Rocky Mountains and near the basin outlet in the southeastern portion of the Missouri River Basin. However, sensitivity to parameterization varies by location. For example, the SFC experiment provided relatively higher KGE in lowland subbasins compared to the default experiment (Fig. 2c). The RUN experiment also shows improved performance in lowland areas as well as in high-elevation subbasins in the western part of the domain (Fig. 2b). The INF experiment produced better simulation results in the northern part of the Missouri River Basin, where grassland dominates (Fig. 2d).

To highlight these differences, we generated ΔKGE maps that display the improvement in KGE for each experiment relative to the default (Fig. 3). The location of improvements differed by experiment. The RUN experiment demonstrated widespread performance improvements across most of the study domain (Fig. 3a). However, the effectiveness of this experiment is diminished in the subbasins located near the outlet of the Missouri River Basin, which is located at the bottom right of the domain. The SFC experiment performance improvements were concentrated in the lower subbasin regions (Fig. 3b). The INF experiment did not exhibit a clear spatial pattern of improvement, with relatively better performance observed in the northern part of Missouri River Basin (Fig. 3c). Finally, the STC experiment showed moderate improvements in subbasins near the outlet of the domain, while substantial improvements were limited (Fig. 3d).

Figure 2 KGE of Noah-MP runoff relative to USGS streamflow at 50 HCDNs. Results are presented for (a) the default setting and with different parameterization scheme of (b) runoff and groundwater (RUN), (c) surface exchange coefficient for heat (SFC), (d) frozen soil permeability (INF) and (e) snow/soil temperature time scheme (STC).

○ No Improvement (^KGE < 0) ● Moderate Improvement (^KGE > 0) ● Substantial Improvement (^KGE > 0.1)

Figure 3 Changes in KGE at 50 HCDNs by the alternative parameterization scheme of (a) runoff and groundwater (RUN), (b) surface exchange coefficient for heat (SFC), (c) frozen soil permeability (INF) and (d) snow/soil temperature time scheme (STC). Subbasins are classified into three categories with different colors indicating No Improvement, Moderate Improvement, and Substantial Improvement.

, 10

To assess how alternative parameterization schemes affect hydrological processes, a comprehensive evaluation of hydrometeorological variables was conducted across multiple representative subbasins. Figure 4 shows precipitation, evapotranspiration, groundwater storage, SWE, runoff time-series for four selected subbasins. These subbasins were selected because they showed notable improvements in KGE values and exhibit process-specific response for each experiment. Their geographic locations are shown in Fig. 1c. The selected subbasins represent diverse hydroclimatic settings: Battle Creek Keystone Basin (warm-summer humid continental, evergreen needleleaf forest, montane forest snow), Gasconade River Basin (humid subtropical, wooded grassland, ephemeral snow), Bear Den Creek Basin (warm-summer humid continental, grassland, prairie snow), and Dinwoody Creek Basin (subarctic, woodland/scrubland, tundra snow).

For the subbasins shown in Fig. 4, the time series comparisons demonstrate how the alternative schemes address limitations of the default setting. The corresponding KGE values indicate improvements based on runoff comparisons with observed streamflow. In Battle Creek Keystone (Fig. 4a), the default shows limited groundwater variability. The RUN experiment produces clear seasonal and interannual groundwater variability which in turn improved simulation of peak runoff in 2014, 2018, and 2023 and increases KGE by 0.194. The 2019 peak was still lower than observed but is more accurately represented than in the default scheme. In the Gasconade River (Fig. 4b), the SFC experiment reduces evapotranspiration relative to the default and raises KGE by 0.372. It corrects the systematic underestimation of flood peaks from 2014–2023, including the largest event in 2017. In Bear Den Creek (Fig. 4c), the default fails to capture the rapid increases in surface runoff, while the INF experiment resolves this limitation. It improves KGE by 0.478 with better simulation of the 2019 and 2023 peaks. In Dinwoody Creek (Fig. 4d), the STC experiment enhances SWE simulation and produces a KGE improvement of 0.142. The alternative SWE aligns more closely with UA SWE and results in more realistic runoff, particularly during the 2015, 2019, and 2023 floods. Collectively, these results show how each experiment resolves distinct weaknesses of the default and improves runoff representation across diverse hydroclimatic conditions.

Figure 4 Time series of precipitation, evapotranspiration, groundwater storage, SWE and total runoff of subbasin with notable improvement of KGE by the alternative scheme of (a) runoff and groundwater (RUN), (b) surface exchange coefficient for heat (SFC), (c) frozen soil permeability (INF) and (d) snow/soil temperature time scheme (STC). Observations include USGS streamflow and UA SWE data. Default results are indicated by dashed lines, and alternative results by solid lines. Precipitation and evapotranspiration panels share the same y-axis.

Figure 4 (continued)

# 375 5.3 Performance Improvement over Land Cover, Climate, and Snow Characteristics

Figure 5 presents the KGE differences between the default and alternative parameterization schemes, illustrating how model performance varies with basin characteristics such as climate, land cover, and snow type. In the Ephemeral Snow class, the RUN and SFC experiments are effective, improving more than 70 percent of the subbasins with average KGE increases of 0.14 and 0.28, respectively. In contrast, INF and STC are not effective, with average increases of only 0.05 and 0.02, likely because adjusting infiltration or snow temperature time schemes does not improve conditions in shallow snow. In the Prairie Snow class, the INF experiment shows some effectiveness, improving more than 80 percent of the subbasins with an average KGE increase of 0.17, likely reflecting differences in soil freezing. The RUN (0.12) and SFC (0.10) experiments also improved performance in more than half of the subbasins. In the Montane Forest and Boreal Forest Snow classes, the baseline model performs best and changes in parameterization typically make conditions worse. For example, while more than 80 percent of the Boreal Forest Snow subbasins improve for the INF experiment the result is only a small increase in KGE of 0.04; the other experiments

are not effective. In the Tundra Snow class, the SFC experiment improves more than half of the subbasins, but the average increase is small at 0.05, so it also has limited effectiveness, while the other experiments show no improvement on average.

**Figure 5** Dot plots showing changes in KGE across basin characteristics. Columns represent (a) climate classification, (b) land cover, and (c) snow classification. Rows represent the RUN experiment (top), SFC experiment (second), INF experiment (third), and STC experiment (bottom). Different experiments are distinguished by symbol colors.

# 5.4 Improvement of Snow Water Equivalent and Snowmelt-Driven Runoff Simulation

While previous results focused on overall runoff evaluation, this section examines the snow component of the Noah-MP model and assesses how alternative schemes impact SWE and snowmelt-driven runoff simulation. The analysis focused on the maximum SWE for each water year and total runoff from April to July (Q), along with their ratio (Q/SWE), as these metrics are important to snowmelt-driven runoff generation (Lapides et al., 2022; Gottlieb and Mankin, 2024). Relative bias was calculated as the difference between the average of simulated values and the average of observed values (simulated minus observed), divided by the average of observed values.

Figure 6 show how the relative bias of SWE varies among subbasins under different parameterization schemes. To examine whether these results relate to SWE variability, we calculated the mean annual maximum SWE for each

HUC8 subbasins in Missouri River Basin over the 2014–2023 period (Fig. S1). Overall, the default, RUN, SFC, and INF experiments produced similar spatial patterns, with systematic underestimation of SWE across most high-snowfall regions in the Rocky Mountains (Fig. S1). In contrast, the STC experiment effectively mitigated this underestimation issue. The number of subbasins with relative bias within ±10 percent (indicated by bold boundaries in Fig. 6) was notably higher for the STC experiment, with 14 subbasins, compared with the default, RUN, SFC, and INF experiments, which had four, four, eight, and six subbasins, respectively. This indicates that while most schemes perform comparably across the basin domain, the STC experiment provides an advantage in snow-dominated subbasin where annual SWE is high and snowmelt-driven runoff is the dominant source of streamflow.

Figure 6 Relative bias of SWE in different HCDNs for (a) default setting and alternative scheme of (b) runoff and groundwater (RUN), (c) surface exchange coefficient for heat (SFC), (d) frozen soil permeability (INF) and (e) snow/soil temperature time scheme (STC). Subbasins with mean bias less than 10% are highlighted using bold boundaries.

The improved representation of SWE led to better Q/SWE ratios in the simulations. To better understand this improvement in SWE generation, we analyzed scatter plots from the STC experiments comparing Q, SWE, and Q/SWE separately (Fig. 7). The analyzed subbasins were selected based on their substantial improvements in the relative bias of Q/SWE, with Q and SWE also considered as reference variables to illustrate their clear responses to the STC experiment. Their geographic locations are illustrated in Fig. 1d. All three subbasins share a subarctic climate but vary in land cover and snow classification: Yellowstone River Basin (subarctic, woodland/scrubland, boreal forest snow), Dinwoody Creek Basin (subarctic, woodland/scrubland, tundra snow), and North Brush Creek Basin (subarctic, evergreen needleleaf forest, montane forest snow).

Figure 7 Scatter plots of SWE (top row), April-July runoff (middle row) and their ratio (bottom row) from subbasin with notable improvement of Q/SWE by the alternative snow and soil temperature time scheme (STC) for (a) Yellowstone River Basin (left column), (b) Dinwoody Creek Basin (middle column), and (c) North Brush Creek Basin (right column). In each panel, the x-axis represents observations and the y-axis represents simulations. Green dots show results from the default experiment, and purple dots show results from the STC experiment. Relative bias for both experiments is shown in the top left of each panel.

All three subbasins show an underestimation of SWE under the default setting, with mean bias values of -28% for Yellowstone River, -44% for Dinwoody Creek, and -19% for North Brush Creek. The STC experiment reduces these biases substantially to -13%, -23%, and +1%, respectively. For April–July runoff (second row), the STC experiment also improves the runoff simulation, with mean bias decreasing from -12% to -4% in Yellowstone, -31% to -26% in Dinwoody, and +9% to +3% in North Brush Creek. The Q/SWE ratio (third row) shows the most consistent reductions in bias, with mean bias improving from +22% to +10% in Yellowstone, +20% to -7% in Dinwoody, and +35% to

+2% in North Brush Creek. Collectively, these results highlight that the STC scheme reduces bias in both SWE and April–July runoff, leading to substantial improvement in Q/SWE across selected subbasins.

#### **6 Discussions**

# 445 6.1 Influences of Alternative Parameterization Scheme on Runoff Generation

This study evaluated the performance of four alternative parameterization schemes in comparison with the default Noah-MP configuration to assess their impacts on runoff generation and other hydrological processes. The analysis revealed three limitations in the default parameterization scheme that were better addressed by the alternative schemes.

The first limitation identified was the inadequate groundwater dynamics in the default groundwater scheme. Groundwater simulated by the default scheme does not always exhibit strong seasonality, with minimal variability throughout the year (Fig. 4a). This deficiency has been previously reported in regions with shallow water tables and was attributed to weak representation of capillary rise (Cai et al., 2014; Wang et al., 2018; Li et al., 2021). Maina et al. (2025) also identified the low capillary rise issue and addressed it by integrating the ParFlow model with Noah-MP. In this study, it was found that the alternative TOPMODEL with equilibrium water table scheme could overcome this limitation to better simulate rapid response of shallow groundwater systems.

The second limitation involved systematic overestimation of ET in vegetated areas, particularly those dominated by cropland. Previous research has demonstrated that the default Noah-MP crop parameterization inadequately represents dynamic cropland processes, leading to possible overestimation of ET (Liu et al., 2016). Furthermore, Noah-MP's consistent overestimation of gross primary productivity in vegetated areas exacerbates this issue by elevating ET in both cropland and grassland environments (Ma et al., 2017). The alternative surface exchange coefficient scheme addresses this limitation by using a different function for calculating surface exchange coefficients for heat transfer. Unlike the default scheme, the alternative approach excludes zero-displacement height considerations and dynamically calculates thermal roughness length. This results in ET estimates that better align with observed values by enhancing the efficiency of heat and moisture exchange between the land surface and atmosphere.

A third limitation is the potential underrepresentation of rapid surface runoff generation that can occur during ROS events. The default Noah-MP configuration inadequately captures the physical processes associated with ROS events, including ice lens formation, changes in snowpack thermal conductivity, and alterations in water retention capacity (Yang et al., 2023; Letcher et al., 2024). The default frozen soil permeability scheme tends to predict more permeable frozen soils than typically observed in natural systems, particularly in conditions where ice layers inhibit infiltration. The alternative scheme overcomes this limitation through a nonlinear weighting function based on volumetric ice content, causing a much larger reduction in diffusivity as ice content increases, which improves the representation of rapid runoff generation during ROS events.

#### 6.2 Recommendations for Parameterization Scheme Selection

Two key criteria were applied in recommending alternative parameterization schemes (Table 3). First, at least 50 percent of the subbasins within each environmental classification had to show improvement relative to the default to ensure sufficient representation. Second, the experiment needed to demonstrate an average KGE improvement of at least 0.1 within that classification.

**Table 3** Recommended Noah-MP parameterization schemes by environmental classification based on two criteria: at least 50% of subbasins within a class showing improvement relative to the default and an average KGE increase of at least 0.1.

| Classification | Categories                       | Recommendation |     |          |     |     |
|----------------|----------------------------------|----------------|-----|----------|-----|-----|
|                |                                  | Default        | RUN | SFC      | INF | STC |
| Climate        | Cold Semi-Arid                   |                | ✓   |          |     |     |
|                | Humid Subtropical                |                |     | ✓        |     |     |
|                | Hot-Summer Humid<br>Continental  |                | ✓   | <b>√</b> |     |     |
|                | Warm-Summer Humid<br>Continental |                | ✓   |          | ✓   |     |
|                | Subarctic                        | ✓              |     |          |     |     |
|                | Evergreen Needleleaf Forest      |                | ✓   |          |     |     |
|                | Woodland/Scrubland               | ✓              |     |          |     |     |
| Land cover     | Wooded Grassland                 |                |     | ✓        |     |     |
|                | Grassland                        |                |     |          | ✓   |     |
|                | Cropland                         |                | ✓   | ✓        |     |     |
| Snow           | Ephemeral                        |                | ✓   | ✓        |     |     |
|                | Prairie                          |                | √   | ✓        | √   |     |
|                | Montane Forest                   | ✓              |     |          |     |     |
|                | Boreal Forest                    | <b>√</b>       | •   | •        |     |     |
|                | Tundra                           | <b>√</b>       | •   | •        |     |     |

The recommendations summarized in Table 3 show that the RUN and SFC experiments are broadly effective across climate, land cover, and snow classifications. The INF experiment shows more limited but targeted improvements, mainly in Warm-Summer Humid Continental, Grassland, and Prairie Snow classifications. In contrast, the STC experiment does not meet the criteria in any classification, and the default is retained for Subarctic climates as well as Woodland/Scrubland, Montane Forest, Boreal Forest, and Tundra environments.

The effects of the recommended parameterization schemes in this study are consistent with previous research. For example, Hamitouche et al. (2025) showed that the TOPMODEL scheme with an equilibrium water table (RUN experiment) can improve model performance by reducing runoff underestimation and yielding a higher KGE than the default setting on a global scale. Chang et al. (2020) reported that the Chen97 surface exchange coefficient scheme (SFC experiment) enhances latent and sensible heat flux simulations, resulting in more accurate evapotranspiration estimates than the Monin-Obukhov scheme over a subtropical forest in southern China. Improvements related to frozen soil processes were also highlighted by Hu et al. (2023), who showed that Koren99's lower soil permeability

(INF experiment) can better represent frozen soil hydrology by capturing stronger soil moisture variability and producing more realistic runoff responses in permafrost regions of the Tibetan Plateau.

#### 500 6.3 The Role of Snow-Related Processes and Implications for Model Selection

An insight from this study is that improvements in overall hydrological performance do not always coincide with better representation of snow processes in Noah-MP. The STC experiment exemplifies this complexity, demonstrating limited improvements in overall runoff simulation (Fig. 3d) while achieving the most substantial enhancements in SWE representation and snowmelt-driven runoff ratios (Fig. 6e). This divergent performance pattern indicates that the STC scheme primarily influences snow accumulation and melt processes rather than the broader suite of hydrological processes that govern total runoff. Similarly, Sthapit et al. (2022) reported that although snowpack dynamics strongly influence runoff and soil moisture, mismatches and temporal lags in the simulated response indicate that improvements in snow representation do not always translate into better streamflow performance.

These findings highlight the importance of considering application needs when selecting parameterization scheme. In snow-dominated basins where water supply forecasting, flood risk assessment, or ecological management depends on snowpack dynamics, the selection of parameterization schemes should prioritize snow process accuracy even when overall runoff metrics show modest improvements. In basins where snow plays a minor role, however, total runoff metrics may remain the most relevant criterion.

# 515 **6.4 Limitation and Future Direction**

While this study advances understanding of parameterization selection in the Noah-MP model, several limitations should be acknowledged regarding the generalizability of our findings. The current analysis is confined to the Missouri River Basin for the period from 2014 to 2023, which may limit the broader applicability of the recommended parameterization scheme. Further validation is essential to establish the robustness of our alternative parameterization approaches across diverse geographical settings to see if the parameterization scheme recommendations by environmental class still hold.

This study focused on a subset of runoff and snow related parameters within the Noah-MP model. Future work should extend the analysis to include additional snow-related processes, such as snow thermal conductivity and snow cover fraction parameterizations. Snow thermal conductivity regulates the transfer of energy through the snowpack and thereby influences the timing of snowmelt, while snow cover fraction controls the representation of sub-grid snow heterogeneity and affects surface energy balance (Yang et al., 2023; Cao et al., 2024; Abolafia-Rosenzweig et al., 2025). Incorporating these processes would enable a more comprehensive assessment of Noah-MP sensitivity to snow dynamics.

The Noah-MP model offers multiple physics parameterization schemes across various land surface processes, where interactions among different schemes can substantially affect model performance. This study focused on individual

parameter sensitivity but did not examine how alternative physics options interact when combined. Future investigations should implement methodologies that evaluate different combinations of physics schemes, thereby delivering a more comprehensive understanding of model uncertainty and overall performance.

In addition to these parameterization schemes, other improvements in model physics may help enhance model performance. This study revealed several key areas where process-based enhancements could address current model limitations, including the underestimation of evapotranspiration and SWE, along with the misrepresentation of rapid surface runoff generation and groundwater dynamics. Incorporating more sophisticated algorithms for snow processes, improving groundwater-surface water interactions, and refining the representation of rapid surface flow processes could substantially enhance the model's ability to simulate realistic hydrological responses.

#### 7 Conclusions

We evaluated the impact of alternative Noah-MP parameterization schemes on runoff and snow generation across 50
USGS HCDN subbasins in the Missouri River Basin. By comparing the default and alternative schemes of runoff and groundwater processes, surface exchange coefficients, frozen soil permeability, and snow/soil temperature time schemes, we identified basin-specific parameterization schemes that provide improved hydrological model performance in snow-dominated watersheds (Figs. 2 and 3). The effectiveness of each scheme varies by basin characteristics including climate, land cover, and snow regime influenced, which supports the use of targeted parameterization schemes instead of relying on default physics (Figs. 5 and Table 2). The STC experiment, while offering limited improvement in overall runoff, produced the greatest increases in snow process representation by reducing SWE bias and improving snowmelt driven runoff dynamics (Figs. 6 and 7).

While this study enhances understanding of parameterization selection in the Noah-MP model, several limitations should be acknowledged. The analysis was confined to the Missouri River Basin for the 2014 to 2023 period, which may limit the broader applicability of the recommended parameterization schemes. This study also focused on a subset of runoff- and snow-related parameters within the Noah-MP model. Future work should extend the analysis to include additional snow-related processes, such as snow thermal conductivity and snow cover fraction parameterizations. Furthermore, evaluating interactions among parameterization schemes will provide a more comprehensive understanding of model uncertainty and performance.

This study provides useful insights into basin-specific parameterization schemes for Noah-MP land surface model applications, with meaningful implications for operational hydrology. The results show that the default configuration remains effective in some environments, such as Subarctic climates and high elevation snow regimes. At the same time, there is potential to improve model performance in other settings by addressing the issues we identified, including overestimation of evapotranspiration, limited representation of frozen soil processes, underestimation of snow dynamics, and weak simulation of groundwater dynamics. In conclusion, these findings suggest that Noah MP

is well tuned in certain conditions, but targeted modifications can substantially enhance its reliability for water supply forecasting and flood risk management across diverse hydroclimatic regions.

### 570 Code Availability

All LIS configuration files, selected basin boundaries, R scripts, and RData files developed in this study are available at Zenodo (Cho et al., 2025; https://zenodo.org/records/17519636; last access: 4 November 2025). The LIS configuration files correspond to the Noah-MP simulations for the default setup and four alternative parameterization schemes (RUN, SFC, INF, and STC). The R scripts and RData files include all materials required to reproduce Figs. 1–7 and ensure full reproducibility of the analysis. The NASA LIS framework, including the Noah-MP land surface model, is freely available at https://github.com/NASA-LIS/LISF (Kumar et al., 2006, 2008; Peters-Lidard et al., 2007).

#### **Data Availability**

The NLDAS-2 meteorological forcing dataset used in this study is available from NASA's Land Data Assimilation Systems website (https://ldas.gsfc.nasa.gov/nldas/v2/forcing; Xia et al., 2012). The basin shapefiles of HCDNs are 580 available **USGS** from the ScienceBase Catalog (https://www.sciencebase.gov/catalog/item/631405bbd34e36012efa304a; Falcone, 2011). The DEM is available from the USGS National Map Download Platform (https://apps.nationalmap.gov/downloader/). Land-cover information is available from the University of Maryland Global Land Cover product (Hansen et al., 2000; https://ldas.gsfc.nasa.gov/nldas/vegetation-class). Climate classification data are available from the Köppen-Geiger climate classification map at 1 km spatial resolution (Beck et al., 2018; https://www.gloh2o.org/koppen/). The snow classification information is available from the Sturm snow classification map (Sturm and Liston, 2021; https://nsidc.org/data/nsidc-0768/versions/1). The observed daily streamflow data are available from the USGS National Water Information System (https://waterdata.usgs.gov/nwis). The UA daily 4 km SWE dataset (version 1) is publicly available at https://nsidc.org/data/nsidc-0719/versions/1 (Broxton et al., 2019).

### 590 Author contributions

EC1 (Eunsaem Cho) conceptualized the research, developed the methodology, and conducted the formal analysis and investigation, as well as prepared the original draft. EC2 (Eunsang Cho) conceptualized the research, contributed to methodology development, and supervised the research throughout the entire research process. CV contributed to the methodology, formal analysis, visualization, and also supervised the research. BL and JMJ contributed to the methodology, formal analysis, and visualization, providing technical and scientific inputs. All authors reviewed and edited the paper.

### Competing interests

The contact author has declared that none of the authors has any competing interests.

### Acknowledgments

The authors further acknowledge the use of computing resources provided by the NASA Center for Climate Simulation.

### **Financial Support**

This research was supported by NASA's Subseasonal-to-Seasonal Hydrometeorological Prediction program (Grant No. 80NSSC24K1278).

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
