# Peer review of "Basin-scale Evaluation of the Noah-MP Land Surface Model for Runoff and Snow Generation in the Missouri River Basin: Insights and Recommendations for Parameterization Scheme Selection"

_EGUsphere, 2025_

## Referee Comment (RC2)

[referee-annotated manuscript omitted]

---

## Author Comment (AC1)

**Responses to Comments from Reviewer #1**

General Comments:

This manuscript provides a detailed evaluation and comparison of runoff and snow simulations using different parameterization schemes of the Noah-MP land surface model in the Missouri River Basin. This work holds scientific value for disaster prevention and mitigation in the basin. However, I have several major concerns, detailed below.

Response: We sincerely thank the reviewer for recognizing the scientific value of our work for disaster prevention and mitigation in the Missouri River Basin and appreciate the thorough review and constructive comments.

   To address the reviewer's comments, we will conduct additional spin-up simulations, as suggested, and revise our conclusions to better acknowledge the limitations of the study, and clarify how this study complement previous studies. We believe these revisions, along with those addressing Reviewer#2's comments, will substantially improve the manuscript. Please see detailed responses below.

Q1. Line 150: I find it unreasonable to use the same initial conditions for the alternative parameterization experiments as for the default simulation. This approach likely affects some of the study's conclusions. I recommend performing a separate spin-up for each experiment. If computational resources are a constraint, the period from January 1, 1995, to December 31, 1999, could be used as the model spin-up phase.

Response: We appreciate your comment. We would like to clarify that analyses presented are for 2014-2023 only. We believe that the preceding 14-year period (2000-2013) should be sufficient for key state variables, including soil moisture and groundwater storage, to reach equilibrium in a wet climate following the initial spin up for 1980-1999. Nevertheless, to address your concern, we will conduct additional spin up simulation as suggested and compare the results with those presented in our initial submission. We will discuss these results in the revised manuscript.

Q2. Section 6.1: The study concludes that the alternative schemes address three specific limitations of the default scheme. I find these conclusions to be overly strong. Previous research suggests that the performance of land surface process schemes is often region-dependent.

Response: We agree that our conclusions should be more carefully framed to reflect the limited hydroclimate conditions of the study area. We will revise Section 6.1 to emphasize that the reported improvements are specific for the Missouri River Basin, and our recommendations may not be applicable in other regions with different hydroclimatic conditions. We will also discuss findings from previous studies to highlight the region dependency of parameterization schemes.

Q3. Section 6.4: I agree with the two limitations identified in this section: first, that the scheme recommendations may not be directly applicable to other regions, and second, that the interactions between different land surface physical processes are not considered. I strongly recommend providing a more in-depth discussion of these points rather than merely noting them in passing.

Response: We appreciate this suggestion. We will expand Section 6.4 to provide more in-depth discussion of the two identified limitations. For transferability, we will include discussions of previous studies that have shown varied performance with these parameterization schemes and how basin-specific characteristics may influence the applicability of our findings to other regions. Similarly, for process interactions, we will incorporate previous findings on coupled parameterization effects to better explain how different schemes may lead to degradation or improvement when combined. We will also discuss recent advances in multi-physics ensemble approaches and their implications for parameterization selection strategies in land surface modeling.

Q4. In Figure 1, does the land cover map represent the actual dataset used in the simulations? Was the IGBP land cover classification scheme employed?

Response: Thank you for your comment. Yes, we used the IGBP land cover classification scheme in our simulations. We will revise Figure 1 and update the results to accurately reflect the land cover dataset used in this study.

Q5. In Figure 4a, the default scheme fails to accurately reproduce the seasonal variation in groundwater storage, whereas it performs well in Figure 4b. Could the authors please attempt to explain this discrepancy?

Response: Previous studies have shown that groundwater estimates from the default scheme do not always show strong seasonal variation (Xia et al., 2017). A major reason is that the scheme has a weak representation of capillary rise, an important force moving groundwater upward for ET consumption during dry seasons and thus may be unable to capture seasonal variation in groundwater in drier regions. To better explain the discrepancy between Figures 4a and 4b, we will examine precipitation minus evapotranspiration (P–ET) as an indicator of basin-scale recharge conditions. We will incorporate this clarification into the revised manuscript to better connect groundwater seasonality with recharge-related controls.

Q6. Is "ROS" in the manuscript an abbreviation for "rain-on-snow"? Please clarify.

Response: Yes, ROS stands for "rain-on-snow." We will define this abbreviation at its first appearance in the manuscript and ensure its consistent usage throughout the manuscript.